# Converging sensory and motor cortical inputs onto the same striatal neurons: An *in vivo* intracellular investigation

Stéphane Charpier[1,2]*, Morgane Pidoux[1], Séverine Mahon[1]

**1** Institut du Cerveau et de la Moelle épinière, ICM, INSERM UMRS 1127, CNRS UMR 7225, Hôpital Pitié-Salpêtrière, Paris, France, **2** Sorbonne Université, UPMC Université Paris 06, Paris, France

* stephane.charpier@upmc.fr

**Data Availability Statement:** All relevant data are within the manuscript.

**Funding:** This research was funded by the Fondation de France, the Fondation pour la Recherche Médicale, the Agence Nationale de la

## Abstract

The striatum is involved in the completion and optimization of sensorimotor tasks. In rodents, its dorsolateral part receives converging glutamatergic corticostriatal (CS) inputs from whisker-related primary somatosensory (S1) and motor (M1) cortical areas, which are interconnected at the cortical level. Although it has been demonstrated that the medium-spiny neurons (MSNs) from the dorsolateral striatum process sensory information from the whiskers *via* the S1 CS pathway, the functional impact of the corresponding M1 CS inputs onto the same striatal neurons remained unknown. Here, by combining *in vivo* S1 electro-corticogram with intracellular recordings from somatosensory MSNs in the rat, we first confirmed the heterogeneity of striatal responsiveness to whisker stimuli, encompassing MSNs responding exclusively by subthreshold synaptic depolarizations, MSNs exhibiting sub- and suprathreshold responses over successive stimulations, and non-responding cells. All recorded MSNs also exhibited clear-cut monosynaptic depolarizing potentials in response to electrical stimulations of the corresponding ipsilateral M1 cortex, which were efficient to fire striatal cells. Since M1-evoked responses in MSNs could result from the intra-cortical recruitment of S1 CS neurons, we performed intracellular recordings of S1 pyramidal neurons and compared their firing latency following M1 stimuli to the latency of striatal synaptic responses. We found that the onset of M1-evoked synaptic responses in MSNs significantly preceded the firing of S1 neurons, demonstrating a direct synaptic excitation of MSNs by M1. However, the firing of MSNs seemed to require the combined excitatory effects of S1 and M1 CS inputs. This study directly demonstrates that the same somatosensory MSNs can process excitatory synaptic inputs from two functionally-related sensory and motor cortical regions converging into the same striatal sector. The effectiveness of these convergent cortical inputs in eliciting action potentials in MSNs may represent a key mechanism of striatum-related sensorimotor behaviors.

Recherche (Grant Number: ANR-16-CE37-0021 2016) and the program 'Investissements d'avenir' ANR-10-IAIHU-06.

**Competing interests:** The authors have declared that no competing interests exist.

## Introduction

The striatum, the main input stage and largest nucleus of the basal ganglia, is involved in the production and selection of adaptive motor behaviors which require the continuous integration of sensory information [1, 2]. To achieve this function, the striatum integrates massive glutamatergic synaptic inputs from a large and diverse set of cortical areas [3, 4]. While corticostriatal (CS) projections obey a topographical organization, each striatal recipient zone receives overlapping convergent inputs from multiple, often functionally-related, cortical regions [2–6].

With respect to its role in the completion of sensorimotor tasks, the dorsolateral part of the striatum processes convergent CS inputs from somatosensory and motor cortical areas [5–10]. The whisker-related CS pathway from the primary somatosensory cortex (S1), which conveys sensory information from the immediate environment to guide rodent exploratory behaviors [11], projects bilaterally to the dorsolateral striatum and forms a clustered pattern of axonal terminals [12, 13], with a few number of postsynaptic contacts per individual afferent CS neurons [14]. S1 CS neurons are distributed uniformly across layer 5 [10, 13, 15, 16], with a significant prevalence of ipsilateral projections [10, 17]. S1 CS inputs are processed by the GABAergic striatal projection neurons—the main ($\sim$90%) neuronal population of the striatum—which have been morphologically defined as medium-sized spiny neurons (MSNs) [18]. We have recently demonstrated *in vivo* that the propagation of whisker-evoked sensory flow through the S1 CS pathway resulted in a selection of sensory information within the striatum [13, 16]. Specifically, while S1 CS neurons reliably responded to the deflection of contralateral whiskers by suprathreshold depolarizing postsynaptic potentials (*d*PSPs), only half of the related MSNs displayed *d*PSPs, which were effective in eliciting action potentials (APs) in one-third of the responding neurons. The remaining striatal neurons did not exhibit any detectable responses to sensory stimuli. This relative inconstancy of striatal responses for a given sensory input was likely due to the patchy organization of S1 CS projections [12, 13] and/or to a powerful shunting synaptic conductance resulting from the feedforward activation of presynaptic striatal GABAergic interneurons by cortical inputs [13, 19–22].

The somatosensory striatal sector also receives ipsilateral synaptic inputs from the CS neurons located in the vibrissal representation of the primary motor cortex (M1), which partially overlap with the S1 CS projections [5, 6, 9, 10, 23–25]. Whisker-related M1 and S1 cortical regions form reciprocal connections [26–28] and converge into the dorsolateral striatum, making this striatal sector an input zone of the sensorimotor corticofugal channel in the basal ganglia. Anatomical studies have revealed the convergence of CS projections from M1 and S1 onto the same striatal GABAergic interneurons [24]. In addition, a recent *in vitro* investigation from the mouse dorsolateral striatum has demonstrated that activation of S1 and M1 CS inputs generated monosynaptic *d*PSPs in both MSNs and striatal interneurons, with an elevated synaptic strength for the M1-induced responses [22]. However, the functional impact of M1 CS projections onto somatosensory MSNs *in vivo*, an experimental condition that fully preserves CS connectivity, has not been investigated so far and we still ignore whether S1 and M1 CS inputs can synaptically-depolarize and fire the same MSN. Addressing these issues is of crucial importance to understand the mechanisms of sensorimotor integration in the striatum. This will also extend our knowledge on the basic processes of integration of cortical information by individual MSNs, which have long been considered as "coincidence detectors" of converging excitatory inputs due to their distinctive electrical membrane properties endowing them with an excessive membrane polarization, a low intrinsic excitability and a weak capacity to summate synaptic events over time from their resting membrane potential [29–32].

In the present study, by combining *in vivo* S1 electrocorticogram (ECoG) with intracellular recordings from somatosensory MSNs and S1 layer 5 pyramidal neurons, we first analyzed the

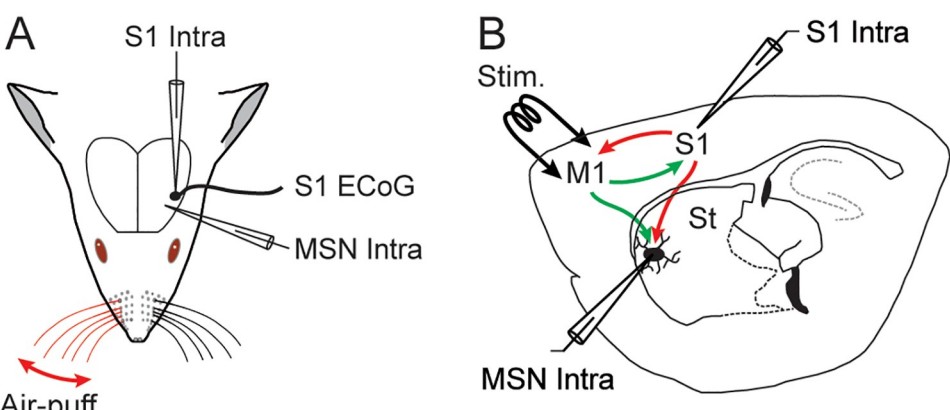

**Fig 1.** *In vivo* **experimental paradigm.** (**A**) Intracellular (Intra) recordings were obtained from the primary somatosensory cortex (S1) and from MSNs located in the S1 projection field of the dorsolateral striatum, simultaneously with an ECoG of S1. Sensory stimuli consisted in iterative air puffs delivered onto the contralateral whiskers and the evoked-responses were collected in the ECoG, S1 neurons and MSNs. (**B**) Activation of the deep layers of the primary orofacial motor cortex (M1), projecting to the somatosensory striatum and to S1 reciprocally, was achieved through low intensity electrical stimulations (Stim.). M1-induced responses were recorded from the somatosensory MSNs and S1 neurons. St: striatum.

responsiveness of these two cell types to deflection of contralateral whiskers (Fig 1A). We then examined their synaptic and firing responses to electrical stimulations of the whisker-related M1 cortex (Fig 1B). By comparing the delay of M1-evoked firing in S1 neurons and the latency of M1-evoked *d*PSPs in the related MSNs, we found that M1 CS pathway could generate short-latency responses in somatosensory MSNs that did not require the intra-cortical recruitment of the S1 CS network (Fig 1B). This study directly demonstrates that the same somatosensory MSNs can process, and be excited by, both somatosensory and motor cortical inputs *in vivo*.

## Materials and methods

The care and experimental manipulation of the animals strictly followed the European Union guidelines (directive 2010/63/EU) and received approval from the French Ministry for Research (# 00773.03) and the Charles Darwin Ethical Committee on Animal Experimentation (C2EA-05). Every precaution was taken to minimize suffering (see general and local anesthesia procedures below) and the number of animals used in each experimental series.

### Animal preparation

Experiments were conducted *in vivo* from 51 adult male Sprague-Dawley rats weighting from 260 to 430 g (Charles River, L'Arbresle, France). Animals were initially anesthetized with sodium pentobarbital (30 mg/kg, i.p.; Centravet, Plancoët, France) and ketamine (25 mg/kg, i.p.; Centravet, Plancoët, France). A cannula was inserted into the trachea and the animal was placed in a stereotaxic frame. Wounds and pressure points were repeatedly (every 2 h) infiltrated with lidocaïne (2%; Centravet, Plancoët, France). After completion of the surgical procedures, moderate doses of pentobarbital (10–15 mg/kg, i.p.) were regularly administered during the recording session and the depth of anesthesia was assessed by continuous monitoring of heart rate and ECoG activity. To obtain long-lasting stable intracellular recordings, rats were immobilized with gallamine triethiodide (40 mg/2h, i.m.; Sigma, France) and artificially ventilated. Body temperature was maintained at 37°C using a feedback-controlled heating blanket. At the end of the experiments, animals received an overdose of sodium pentobarbital (200 mg/kg, i.p.).

## Electrophysiological recordings and stimulations

Spontaneous ECoG activity and cortical sensory-evoked potentials (Fig 1), elicited by forced deflections of contralateral whiskers (see below), were captured with a low impedance ($\approx$ 60 k$\Omega$) silver electrode apposed on the dura above S1 (-2 mm anterior to the bregma; 5 mm lateral to the midline) [33]. The reference electrode was placed on a temporal muscle of the opposite side of the head.

Intracellular recordings from MSNs ($n$ = 76) and S1 pyramidal neurons ($n$ = 11), performed with glass micropipettes filled with 2 M KAc (50–80 M$\Omega$), were obtained in current-clamp mode (Axoclamp-2B amplifier, Axon Instruments, Union City, CA) and in conjunction with the monitoring of S1 ECoG activity (Fig 1A). For intracellular labeling of MSNs, 1% neurobiotin (Vector Laboratories, Burlingame, CA) was added to the pipette solution. MSNs, identified on the basis of their distinctive electrophysiological and morphological properties (Fig 2), were recorded in the ipsilateral striatal projection field of S1 [10, 12, 13, 16], at the corresponding stereotaxic coordinates: -2.4 to -2.1 mm anterior to the bregma, 4.5 to 5.2 mm lateral to the midline, and 3.2 to 5.6 mm ventral to the brain surface [33]. S1 layer 5 pyramidal neurons

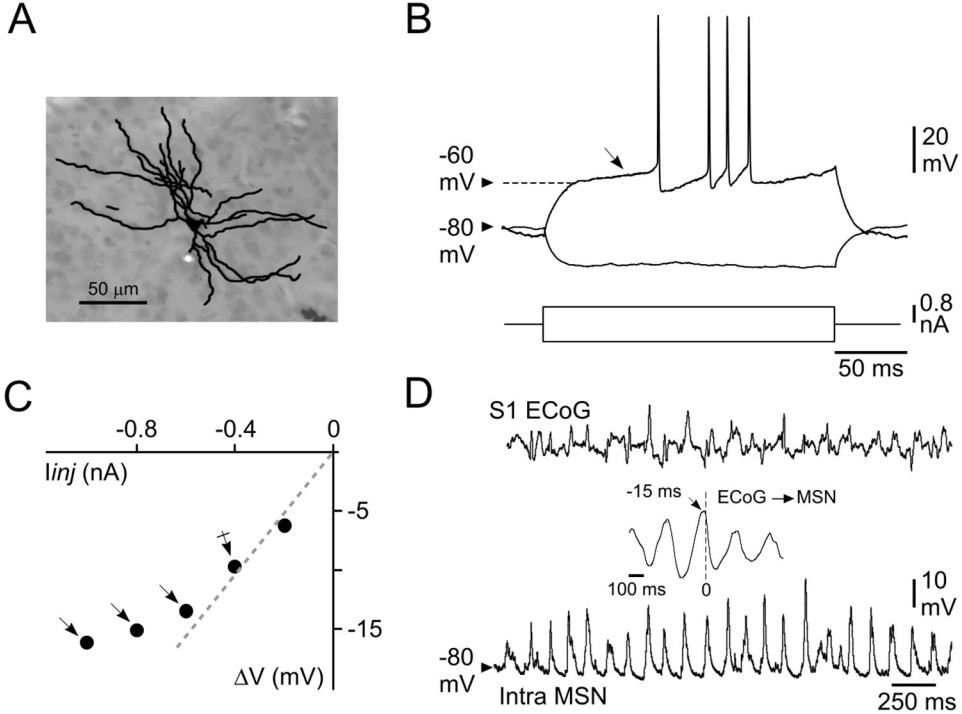

**Fig 2. Morphological and basic electrophysiological properties of somatosensory striatal MSNs.** (**A**) Synthetic representation of a striatal MSN labeled by intracellular injection of neurobiotin. (**B**) Voltage changes and firing patterns (top records) of a MSN in response to the injection of positive (single response) and negative (averaging of 20 trials) current pulses (bottom traces). Note the high membrane polarization at rest and the slow ramp depolarization (from -60 mV, oblique arrow) induced by the depolarizing current pulse, resulting in a delayed firing. (**C**) Plot of voltage changes ($\Delta$V) as a function of the injected current (I$inj$). Note the membrane inward rectification in response to hyperpolarizing currents higher than -0.4 nA (arrows). The dashed line indicates the theoretical linear ohmic relation. R$m$ of MSNs (24 M$\Omega$ for this cell) was thus calculated from the mean voltage response obtained at -0.4 nA (crossed arrow). (**D**) Simultaneous recordings of S1 ECoG (top) and intracellular activity of a MSN (bottom) located in the corresponding CS projection field. The cross-correlation between intracellular and ECoG signals (inset) shows that striatal V$m$ fluctuations closely followed the slow (~5 Hz) cortical oscillation by about 15 ms. Data shown in (**A–D**) are from the same neuron. Here and in the following figures the value of V$m$ is indicated at the left of the intracellular records.

were recorded from the vicinity (< 200 μm) of the ECoG electrode, within the cortical sector projecting to the recorded striatal region [10, 12, 13]: from -2.1 to -1.7 mm anterior to bregma; 4.8 to 5.1 mm lateral to the midline, and 1.2 to 2.1 mm under the cortical surface [33].

Sensory stimulations consisted in puffs (50 ms duration) of compressed air, delivered by a picospritzer unit (Picospritzer III, Intracel LTD, Royston Herts, UK) to the whiskers contralateral to the recording sites through a 1-mm diameter glass pipette (Fig 1A). Air-puffs were given 40 times with a low frequency (0.24 Hz) to prevent adaptation of whisker-evoked responses [34] and activity-dependent intrinsic plasticity in S1 pyramidal neurons [35]. The intensity of stimuli was set at the beginning of each experiment as the minimal air-puff pressure (20–50 psi) generating the largest ECoG response. Under these conditions, the air-puff stimuli deflected 4–8 whiskers by ~10 deg [13, 35].

Intracortical electrical stimulations (200 μs duration) were given 40 times through a bipolar stimulating electrode (1 mm tip separation) inserted (1.7 mm under the brain surface) in the orofacial part of the primary motor cortex (M1, 1.7–2.2 mm anterior to bregma; 1.8–2.2 mm lateral to the midline) [33] (Fig 1B), which is reciprocally connected with S1 [26–28] and whose CS projections overlap with S1 CS projections within the dorsolateral striatum [5, 6, 9, 10, 24]. Cortical stimulations were applied at low intensity (5–25 V) with the same frequency used for sensory stimuli (0.24 Hz). Before curarization of the animals, we ensured that electrical stimulations of M1 could induce movements of one or more whiskers (≈ 3 mm deflection).

## Data acquisition and analysis

Electrophysiological recordings were digitized at 25 kHz for intracellular signals and at 3 kHz for ECoG waveforms (CED 1401plus, Spike2 software version 7.06; Cambridge Electronic Design, Cambridge, UK). Cross-correlograms between subthreshold striatal intracellular activity (1–5 s of continuous recording down-sampled at 3 kHz) and the corresponding ECoG signal (taken as the temporal reference) were computed using Spike2 software. The value of neuronal membrane potential (V$m$) was calculated as the mean of the distribution of spontaneous subthreshold activity (10 s duration). The corresponding magnitude of V$m$ fluctuations was quantified as the standard deviation of the distribution (SD V$m$). V$m$ values were eventually corrected when a tip potential was recorded after termination of the intracellular recording. Measurements of apparent membrane input resistance (R$m$) and time constant (τ$m$) were based on the linear electrical cable theory applied to an idealized isopotential neuron [36]. Because MSNs display prominent inward membrane rectification during injection of large amplitude negative current pulses [31, 32, 37] (Fig 2C), their R$m$ was assessed from the mean membrane voltage drop induced by repeated square current pulses of low intensity (-0.4 nA, $n$ = 20, 200 ms duration, applied every 1.25 sec) (Fig 4D). τ$m$ was derived from an exponential fit applied to the initial part of the current-induced membrane hyperpolarization.

The amplitude of APs was calculated as the voltage difference between their voltage threshold, measured as the membrane potential at which the dV/dt exceeded 10 V/s [35], and the peak of the waveform. Their total duration was measured as the time between their voltage threshold and the return to the same V$m$ value. The spontaneous firing rate of neurons was calculated from a continuous recording period ≥ 10 s.

The latency of intracellular S1- and M1-evoked responses was calculated as the time delay between the stimulus onset (air-puff or cortical electrical stimulus) and the foot of the neuronal responses (Fig 4E). Neuronal events having shape (rising and decay phases) and/or latency that did not match those of the mean synaptic response obtained after averaging of 20–40 trials were not included in the analysis. Amplitude of individual S1- or M1-evoked subthreshold

potential was measured as the voltage difference between the foot and the peak of the response (Fig 4E). The firing probability of striatal and cortical neurons in response to whiskers deflection or M1 activation was calculated as the ratio between the number of suprathreshold synaptic responses and the total number of trials. Firing latency was the time between the stimulus onset and the peak of the AP waveform.

Numerical values are given as means ± standard error of the mean (SEM). Statistical significance was assessed using, appropriately, unpaired Student's *t* tests, one-way analysis of variance or Mann-Whitney Rank Sum test. Statistical analyses were performed with SigmaStat 3.5 (SPSS Inc., Chicago, IL, USA).

## Results

### Morpho-functional properties of somatosensory MSNs and sensory-evoked responses

*In viv*o intracellular recordings were obtained from 76 MSNs located in the striatal region processing somatosensory information originating from the whisker-related S1 (Fig 1). Labeled cells (*n* = 4) displayed the distinctive morphological features of MSNs [13, 18], including somata with diameters between 10 and 20 μm which gave rise to 4–6 ramified primary dendrites densely covered with spines apart from their most proximal region (Fig 2A). Consistently, all recorded striatal neurons also displayed the characteristic electrophysiological properties of MSNs [13, 16, 31, 32, 37]. They exhibited a highly polarized mean V*m* (-79.2 ± 0.5 mV, *n* = 76 MSNs), as compared to pyramidal cortical neurons (see Fig 5A and 5B), associated with a relatively low R*m* (27.1 ± 0.8 MΩ) and a short τ*m* (6.7 ± 0.2 ms) (Fig 2B). Threshold positive current pulses evoked in these cells a ramp-like membrane depolarization, starting at about -60 mV, which led to a long delay in the first AP discharge (Fig 2B). APs, measured on current-induced depolarizations, had an amplitude of 59.7 mV (± 0.8 mV, *n* = 73 MSNs), a total duration of 1.30 ms (± 0.03 ms) and a voltage threshold of -49.8 mV (± 0.2 mV). The membrane current-voltage relationship of recorded neurons consistently displayed a marked inward rectification in response to negative current pulses of increasing intensity (Fig 2C). The typical spontaneous activity of MSNs consisted in a continuous barrage of synaptic depolarizations (Figs 2D and 3, *bottom records*), responsible for large V*m* fluctuations (SD V*m* = 4.5 ± 0.2 mV, *n* = 76 MSNs), causing, however, a low firing rate (0.1–2 Hz) in a restricted number of neurons (15 out of 76). This ongoing rhythmic synaptic activity was tightly correlated with the oscillatory waveforms simultaneously recorded in S1 ECoG, with a negative temporal shift (10–17 ms) (Fig 2D, *inset*) consistent with the unidirectional and monosynaptic excitation of somatosensory MSNs by S1 [13].

We further characterized the functional properties of recorded MSNs on the basis of their responsiveness to sensory stimulations (Fig 1A). The vast majority of MSNs (*n* = 67 of the 76 tested neurons; ∼87%) responded to air-puff deflections (Fig 3), as attested by the clear-cut average sensory responses which exhibited the typical profile of whisker-evoked depolarizing post-synaptic potentials (*d*PSPs) (Fig 3A and 3B) [13]. Among these responding neurons (*n* = 67), 56 displayed only subthreshold *d*PSPs (*d*PSP-responding MSNs) regardless of the intensity of the stimulation (Fig 3A). These subthreshold synaptic responses had a mean latency of 22.5 ± 0.6 ms (*n* = 56 MSNs) and a maximal amplitude of 7.6 ± 0.4 mV. In this group of striatal cells, whisker stimuli were able to generate definite synaptic responses with a probability as high as 0.75 ± 0.4 (*n* = 56 MSNs), demonstrating the robustness of sensory information propagation from the periphery to the dorsal striatum.

Eleven of the 67 responding MSNs could elicit an AP on the sensory-induced *d*PSP (AP-responding MSNs), with however a relatively low probability of evoked discharge, from 0.03 to

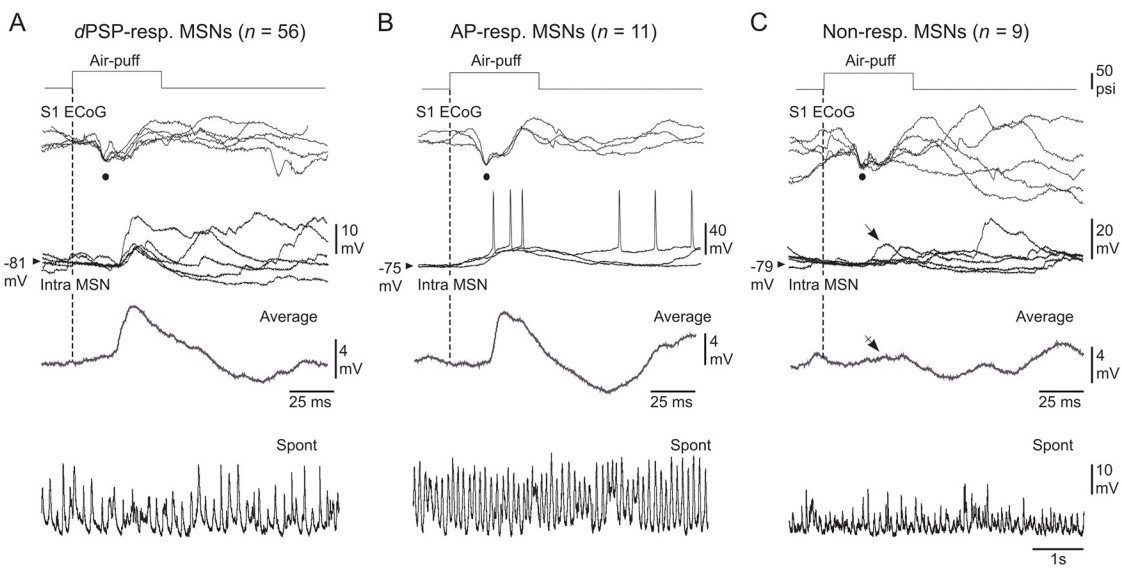

**Fig 3. Characterization of somatosensory MSNs as a function of their responsiveness to a given whisker stimulation.** (**A–C**) Air-puff stimuli of same intensity (optimal stimuli, top traces) induced an early negative deflection in the ECoG (black dots), associated with either subthreshold synaptic responses (**A**, *d*PSP-responding), suprathreshold synaptic responses (**B**, AP-responding) or a lack of synaptic response (**C**, Non-responding) in the simultaneously recorded MSNs (Intra MSN). Three to 5 five successive responses are superimposed. The vertical dashed lines mark the onset of whisker stimuli. The average traces (excluding firing responses) were computed from at least 20 trials. The oblique arrow in (**C**) indicates the occurrence of occasional temporally-matching sensory responses, which was largely curtailed after averaging of 40 successive trials (crossed arrow). The lowest records are segments of spontaneous activity recorded in the corresponding MSNs. Results shown in (**A**), (**B**) and (**C**) are from three distinct experiments.

0.26 ($P_{firing}$ = 0.11 ± 0.02, $n$ = 11 MSNs) (Fig 3B). The latency of APs on whisker-evoked sensory responses was 29.5 ± 1.5 ms ($n$ = 11 MSNs). The amplitude of subthreshold responses in these AP-responding MSNs (12.1 ± 1.9 mV, $n$ = 11 MSNs) was larger than that measured in *d*PSP-responding neurons ($P < 0.001$) and their probability of occurrence over repeated sensory stimulations was increased ($P_{dPSPs}$ = 0.89 ± 0.02, $n$ = 11 MSNs; $P < 0.05$). Voltage firing threshold, as well as R*m* values, calculated from *d*PSPs-responding (-49.7 ± 0.2 mV and 26.8 ± 0.95 MΩ, respectively, $n$ = 56) and AP-responding (-50.5 ± 0.2 mV and 25.4 ± 2.0 MΩ, respectively, $n$ = 11) MSNs, were not significantly different ($P > 0.1$ for both parameters), indicating that the differences in sensory responsiveness did not result from a dissimilar intrinsic excitability between the two striatal cell populations. However, we found that AP-responding neurons had a more depolarized V*m* (-75.4 ± 4 mV, $n$ = 11 AP-responding MSNs *versus* -79.6 ± 0.6 mV, $n$ = 56 *d*PSP-responding MSNs; $P < 0.01$) and larger baseline V*m* fluctuations (SD V*m* = 4.5 ± 0.6 mV, $n$ = 11 AP-responding MSNs *versus* 3.4 ± 0.2 mV, $n$ = 56 *d*PSP-responding MSNs; $P < 0.05$) (Fig 3A and 3B), suggesting a more intense background synaptic activity in the MSNs that fired AP in response to sensory stimulations.

In the 9 remaining MSNs, no sensory-evoked *d*PSP could be detected (Non-responding MSNs) (Fig 3C). Although occasional large amplitude synaptic events could occur within the appropriate temporal window after air-puff stimulations (Fig 3C, *arrow*), they were likely spontaneously generated since they were barely detectable after averaging of a large number of trials (Fig 3C, *average trace*). The R*m* of non-responding neurons (31.1 ± 2.8 MΩ, $n$ = 9) was similar to that of *d*PSP- and AP-responding cells ($P > 0.1$). Their V*m* (-81.0 ± 1.6 mV) was comparable to that measured from MSNs exhibiting subthreshold responses ($P > 0.05$), but was more polarized ($P < 0.05$) and less fluctuating (SD V*m* = 3.8 ± 0.4 mV, $n$ = 9 non-responding MSNs; $P < 0.01$) compared to AP-responding cells (Fig 3C, *bottom trace*).

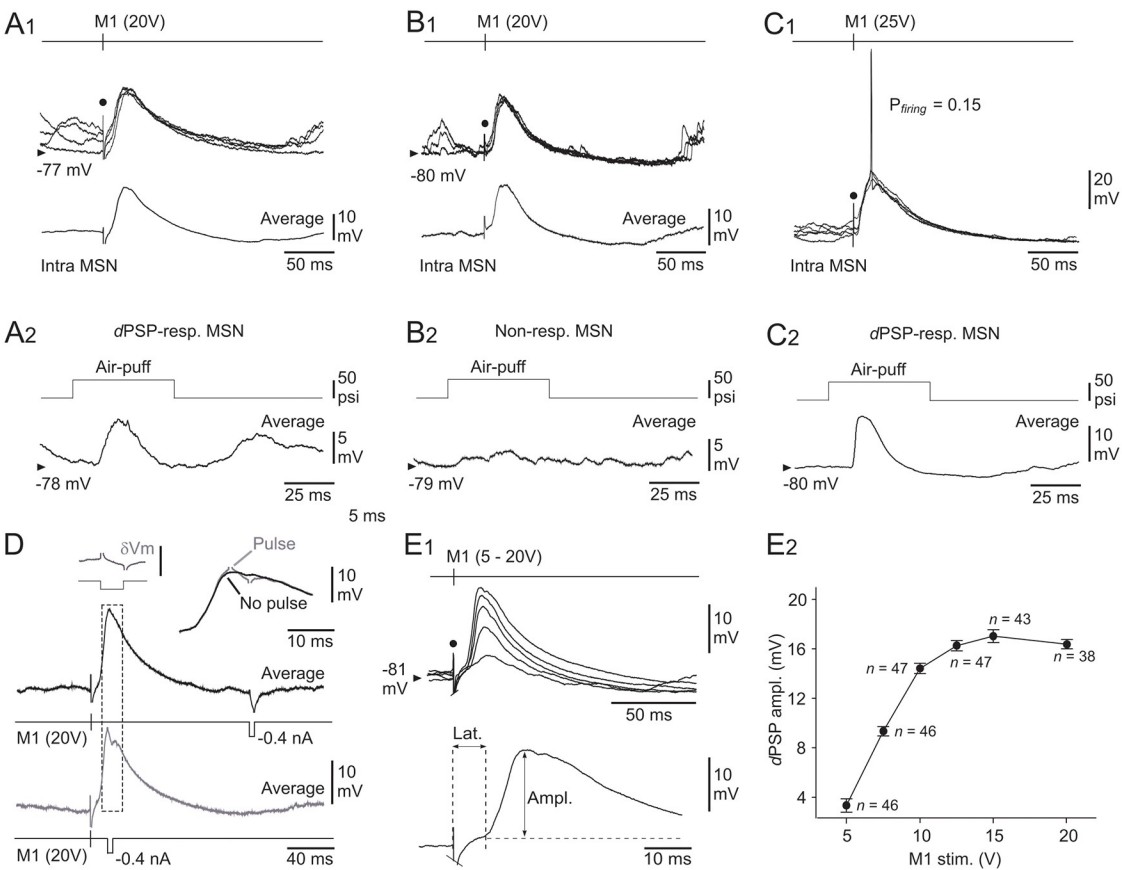

**Fig 4. Properties of M1-induced responses in somatosensory MSNs.** (**A–C**) Intracellular responses (four superimposed records) from three somatosensory MSNs to electrical stimulations of M1 (timing and intensity of stimuli are indicated) (**A1–C1**), and corresponding air-puff induced responses (averaging of 15–40 successive trials) recorded from the same cells (**A2–C2**). The modality of sensory responsiveness in the illustrated neurons is indicated (see text for details). The bottom traces in **A1** and **B1** represent the averaging of at least 20 M1-evoked intracellular potentials. (**D**) Conductance increase upon M1-evoked depolarizations in MSNs. M1-induced potentials (averages of 20 successive responses) in a MSN, coupled with injection of current pulses (-0.4 nA, 5 ms) applied 140 ms after M1 stimulus (black record) or at the peak of the evoked response (grey record). The traces are superimposed and expanded at right. The inset at left ($\delta$V$m$) shows the subtraction of both records (as delimited by the dashed box), demonstrating the collapse of current-induced voltage drop at the peak of the induced depolarization (calibration bar, 5 mV). (**E**) Input-output relation in the M1-somatosensory MSNs circuit. (**E1**) Superimposed synaptic depolarizations (averages of at least 20 responses) evoked in a MSN by M1 stimulations applied with increasing intensity (5 to 20 V). The bottom trace indicates the method used to measure the latency (Lat.) and the amplitude (Ampl.) of the M1-evoked intracellular responses. (**E2**) Pooled data showing the changes in $d$PSPs amplitude in response to M1 stimuli of increasing intensity (M1 stim.). Only MSNs tested with at least two M1 stimulus intensities were computed. Data points correspond to mean ± SEM (the corresponding number of MSNs is indicated). Note the saturation of $d$PSP amplitude from an intensity of 15 V. The black dots in **A–C** and **E1** indicate the electrical stimulus artifact.

Altogether, these findings confirm and extend our previous categorization of somatosensory MSNs [13] as $d$PSP-, AP- and non-responding neurons for a given sensory stimulus.

## Motor cortex-induced synaptic responses in somatosensory MSNs

We next explored, for the first time, the responsiveness of somatosensory MSNs to electrical stimulations (5–25 V, 0.24 Hz) of the orofacial part of M1 known to send CS projections into the somatosensory striatal sector [5, 6, 9, 10, 24]. All tested MSNs ($n$ = 75), previously challenged with sensory stimulations, displayed clear-cut sub- or suprathreshold synaptic depolarizations in response to electrical stimulations of M1.

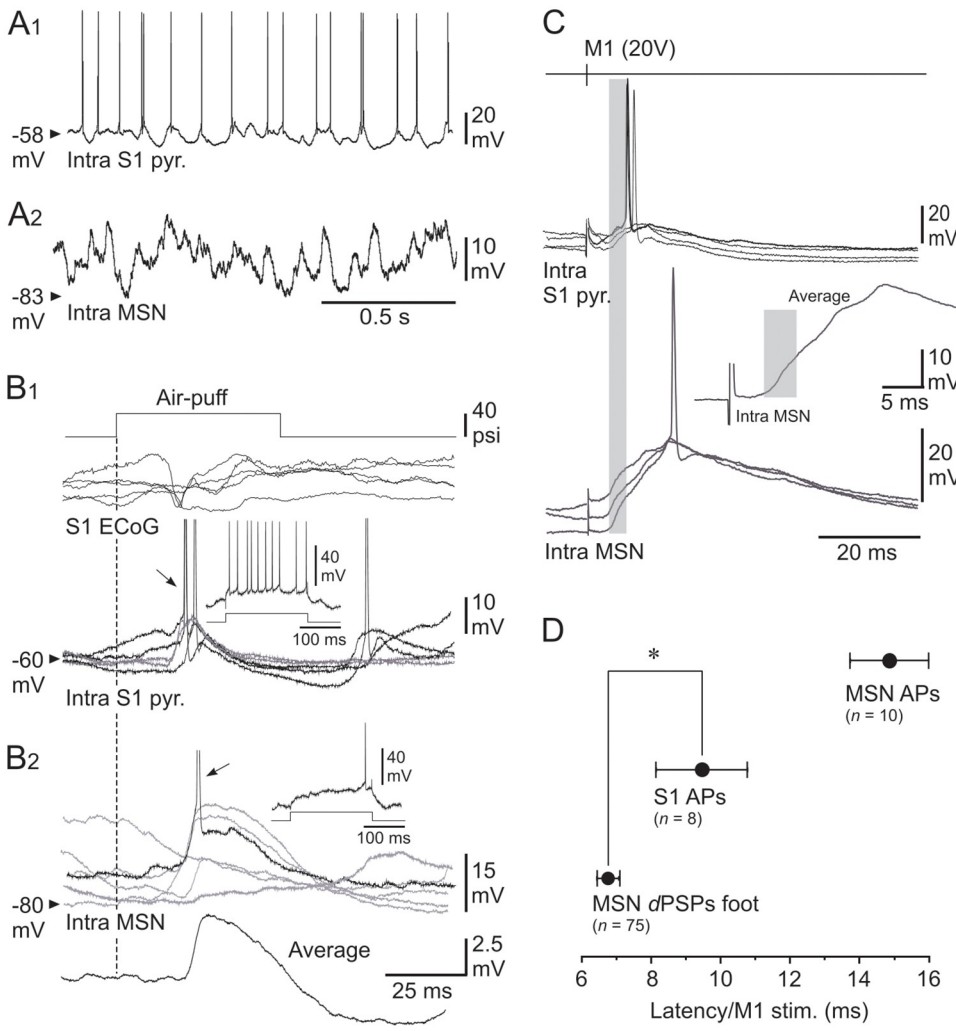

**Fig 5. Functional properties of S1 neurons and their role in the M1-evoked striatal responses.** (**A–B**) Spontaneous activity and sensory-induced responses in the S1 CS pathway. (**A**) Ongoing synaptic and firing activity in a S1 pyramidal neuron (**A1**) and a MSN (**A2**) recorded separately from the same experiment. (**B**) Superimposed ($n = 5$) whisker-induced (top trace, 40 psi) subthreshold (grey records) and suprathreshold (black records) synaptic responses captured from the S1 cell (**B1**) and the MSN (**B2**) shown in (**A**). The arrows indicate the firing (APs are truncated for commodity) caused by sensory stimuli. The *insets* show the cell responses to intracellular application of a current pulse of +0.4 nA (bottom trace). The lowest trace in (**B2**) is the average of 20 successive sensory responses recorded in the MSN. The vertical dashed line indicates the onset of sensory stimuli. (**C**) Superimposed intracellular responses, obtained in a S1 pyramidal neuron (top records) and in a somatosensory MSN (bottom records) from the same rat, to electrical stimulations of M1 with the indicated intensity. The inset is an expanded average ($n = 10$ trials) response recorded from the MSN. The grey boxes cover the initial portion of the M1-induced striatal *d*PSPs that preceded the activation of the S1 neuron. (**D**) Pooled data showing the latency of striatal *d*PSPs foot, striatal firing (MSN APs) and firing of S1 neurons (S1 APs) induced by electrical stimulations (20 V) of M1. The number of computed neurons is indicated. * $P < 0.05$.

Striatal cells which responded to M1 electrical stimulations by subthreshold *d*PSPs ($n = 65$), regardless of the intensity tested, encompassed the three categories of MSNs previously classi-fied by their responsiveness to whisker stimulations: AP-responding ($n = 8$), *d*PSP-responding ($n = 49$) and non-responding MSNs ($n = 8$) (Fig 4A and 4B). These cells had a mean V$m$ and R$m$ of -79.8 ± 0.5 mV and 26.7 ± 0.9 MΩ ($n = 65$), respectively, and a voltage firing threshold of -49.6 ± 0.3 mV ($n = 62$). The onset latency of subthreshold *d*PSPs, calculated from the

maximal stimulus intensity used in each tested cell (15–25V), was 6.7 ± 0.3 ms ($n$ = 65), a value compatible with a monosynaptic CS-induced response in MSNs (see Discussion). The amplitude of M1-induced synaptic responses, which was on average of 15.6 ± 0.8 mV ($n$ = 65) (Fig 4E1 and 4E2), exhibited a relatively large trial-to-trial variability (SD $d$PSPs = 3.5 ± 0.2 mV, $n$ = 65).

In a smaller subset of MSNs ($n$ = 10 out 75), including $d$PSP-responding ($n$ = 6) (Fig 4C), AP-responding ($n$ = 3) and non-responding ($n$ = 1) MSNs to sensory stimuli, electrical stimulation of M1 (15–25V) elicited $d$PSPs that could reach threshold for AP firing (Fig 4C1). The mean V$m$ of these cells was -78.1 ± 2.2 mV ($n$ = 10), a value similar to that measured for MSNs responding to M1 stimuli by merely subthreshold $d$PSPs ($P$ > 0.2). Their R$m$ (28.9 ± 3.1 MΩ), as well as their firing threshold (-50.0 ± 0.6 mV), were also comparable ($P$ > 0.2 for both parameters), indicating that the differential responsiveness of MSNs to M1 stimuli was not caused by a distinct intrinsic excitability. In these MSNs, the suprathreshold responses occurred with a probability of 0.5 ± 0.1 (from 0.005 to 1, $n$ = 10) and were interposed with large amplitude $d$PSPs (18.9 ± 3.1 mV, $n$ = 9). APs latency on M1-induced synaptic depolarizations was 14.9 ± 1.1 ms ($n$ = 10), a value considerably shorter compared to that measured in response to sensory stimulations (29.5 ±1.5, $n$ = 11 MSNs; $P$ < 0.001).

The M1-evoked responses in MSNs could originate from the activation of postsynaptic glutamatergic receptors through the activation of CS neurons, and/or from feedforward GABAergic synaptic inputs caused by the excitation of striatal interneurons by M1 CS projections [20–22, 24]. It is expected that such synaptic inputs would considerably increase MSNs membrane conductance as the M1-induced response develops. We thus compared in 8 MSNs the current-induced (-0.4 nA, 5–20 ms of duration, $n$ = 20 trials) voltage changes generated at the summit of the cell responses (Fig 4D, *grey records*) to those obtained 120–160 ms after M1 stimulations, *i.e.* after termination of synaptic potentials (Fig 4D, *black records*). While the mean voltage drop obtained in between M1-evoked responses was of 8.1 ± 1.2 mV (from 4.6 to 12.4 mV, $n$ = 8 MSNs) (Fig 4D, *black records*), it was entirely cancelled in 4 MSNs at the peak of the evoked depolarization or had a net amplitude, after subtracting the M1-induced potential free of current pulses, less than 1.5 mV ($n$ = 4 MSNs) (Fig 4D, *insets*). These findings demonstrate a dramatic increase in membrane conductance at the time of M1-evoked response, as expected from a mechanism involving the activation of postsynaptic receptors.

Pooling the measurements obtained from MSNs for which at least two intensities of stimulation were tested ($n$ = 56), we found that $d$PSPs increased in amplitude as the intensity of M1 electrical stimuli was augmented (Fig 4E1), indicating a progressive recruitment of excitatory inputs onto the recorded cells. The amplitude of synaptic responses reached saturation (~16 mV) for intensities ≥ 15 V (Fig 4E2), suggesting that nearly all CS neurons that could be stimulated were already activated from this stimulation intensity (see Discussion).

## Activation of S1 pyramidal neurons by M1 stimulations and its relation to striatal responses

The synaptic responses recorded in somatosensory MSNs following electrical stimulation of M1 could primarily result from the intra-cortical synaptic activation of S1 CS neurons, *via* the reciprocal excitatory connections between the two cortical areas [26–28] and/or from an antidromic activation of S1 neurons projecting to M1. To test for these possibilities, we made intracellular recordings from S1 layer 5 pyramidal neurons ($n$ = 12) (Figs 1B, 5A1 and 5B1), known to provide the main CS source to the corresponding ipsilateral striatal sector [10, 13], and sought to determine whether: 1) S1 neurons were synaptically-excited by whiskers deflection (Figs 1A and 5B1), 2) S1 neurons could be synaptically- and/or antidromically-activated by M1 stimuli

(Figs 1B and 5C, *upper traces*) and, if so, 3) their firing upon M1 stimuli could fully account, or not, for the postsynaptic responses recorded in somatosensory MSNs (Fig 5C and 5D).

S1 layer 5 neurons exhibited oscillatory synaptic depolarizations around a mean V$m$ of -65.9 ± 1.5 mV ($n$ = 12). This sustained background synaptic activity resulted in spontaneous firing activity in 10 out of the 12 recorded cells (2.4 ± 0.7 Hz) (Fig 5A1), contrasting with the subthreshold synaptic barrage recorded from MSNs in the same experiments (Fig 5A2). S1 neurons were identified as pyramidal cells on the basis of their intrinsic firing profile in response to suprathreshold direct stimulations, classified as either regular spiking ($n$ = 8) (Fig 5B1, *inset*) or intrinsic bursting ($n$ = 4). In response to sensory stimuli, 11 cortical neurons displayed short latency (15 ± 0.5 ms) synaptic depolarizations (Fig 5B1), which could elicit an AP in the majority of cells with a mean probability of 0.4 ± 0.1 ($n$ = 8) (Fig 5B1). One of the recorded cells exhibited a pure hyperpolarizing postsynaptic potential (15.7 ms of latency) following whiskers deflection. Altogether, these properties are consistent with those already described for whisker-related synaptic responses in layer 5 S1 neurons [35, 38, 39], including pyramidal cells identified as CS neurons [13].

Eleven of the 12 tested S1 pyramidal cells displayed synaptic depolarizations in response to optimal M1 electrical stimulations (15–20 V), with a foot latency of 3.4 ± 0.4 ms. In 3 neurons, the M1-evoked synaptic depolarizations remained subthreshold, with a maximum amplitude of 5.1 ± 1 mV. In the majority of cells (8 out of 12), the M1-evoked synaptic depolarizations reached a maximal amplitude of 8.7 ± 0.9 mV (Fig 5C) and could cause firing with a mean probability of 0.5 ± 0.1 and an AP latency of 9.5 ± 1.3 ms ($n$ = 8 S1 neurons) (Fig 5C and 5D). None of the recorded neurons were antidromically activated, or directly fired, by M1 electrical stimulations in the range of stimuli tested in this study.

As exemplified by the experiment shown in Fig 5C, in which a S1 pyramidal cell (*top records*) and a somatosensory MSN (*bottom records*) were recorded in the same animal, the onset of *d*PSPs evoked in the striatal cell by electrical stimulations of M1 preceded by 4 ms (Fig 5C, *inset*) the initial discharge of S1 neurons induced by the same stimuli. This temporal advance of M1-induced synaptic responses in MSNs relative to S1 neurons firing was significant over the whole set of tested neurons (MSNs *d*PSPs: 6.7 ± 0.3 ms, $n$ = 75 *versus* AP S1 neurons: 9.5 ± 1.3 ms, $n$ = 8 neurons; $P$ < 0.05) (Fig 5D). These findings indicate that M1 stimulations induced a synaptically-mediated activation of S1 pyramidal neurons, instead of direct/antidromic excitation, but that the early synaptic effect of M1 onto somatosensory MSNs was not the result of an intra-cortical recruitment of S1 CS neurons (see Discussion).

## Discussion

### Properties of responses evoked in somatosensory MSNs by M1 stimulations

It is well established that CS projections from M1 and S1 whisker regions converge with a substantial overlap into the dorsolateral striatum [5, 6, 9, 10, 23]. However, the functional impact of excitatory M1 CS inputs onto the MSNs that process sensory information from the related S1 region remained to be characterized. We demonstrated in the present *in vivo* work that electrical stimulation of M1 caused depolarizing synaptic responses in all recorded MSNs, including those that did not display a detectable sensory response. MSNs were here recorded without separating them into direct (striatonigral) and indirect (striatopallidal) pathway neurons [40]. Previous works have shown that both types of striatal neurons could respond to the activation of S1 CS inputs, with however possible differences in their response properties [19, 22, 41]. Given the importance of these two projection pathways in basal ganglia function [42], it will be crucial to investigate in the future whether sensorimotor integration differs between these subpopulations of MSNs.

The M1-evoked potentials in somatosensory MSNs had the properties of monosynaptic *d*PSPs, demonstrating direct and functional excitatory connections from M1 onto ipsilateral striatal neurons. First, striatal responses exhibited the classical shape and duration of postsynaptic potentials, with an onset latency (about 6 ms for optimal stimuli) consistent with a CS monosynaptic delay [43–45] and the conduction velocity of APs in M1 CS axons [46]. Second, the M1-induced responses in MSNs fluctuated in amplitude from trial-to-trial, in agreement with the probabilistic nature of synaptic transmission. Third, the evoked-depolarizations were associated with a dramatic increase in membrane conductance as expected for cortically-generated excitatory synaptic potentials in MSNs [47]. This increase in synaptic conductance was likely aggravated by an additional feedforward GABAergic inhibition of MSNs caused by the activation of striatal interneurons by CS M1 projections [20, 22, 24, 48]. The initial part (the first 3 ms) of M1-evoked *d*PSPs in MSNs probably resulted from the activation of M1 CS neurons and not from the secondary intra-cortical recruitment of S1 CS neurons by electrical stimulation of M1. This conclusion is supported by: 1) the monosynaptic nature of the synaptic responses (see above), 2) the fact that S1 CS neurons were never antidromically-activated in our experimental conditions, a process that could potentially generate a fast response in MSNs *via* an axon reflex in S1 neurons projecting to the striatum and, 3) the M1-induced firing latency of CS S1 pyramidal neurons that consistently, and significantly, followed the early part of the synaptic depolarization in MSNs. It is reasonable to think that the subsequent part of the M1-evoked synaptic depolarization in MSNs represents the successive, and overlapping, effects of M1 and S1 CS neurons, the latter being monosynaptically-activated by the intra-cortical excitatory projections from the whisker region of M1 [26]. Such a process is in agreement with the short-latency (~3 ms) synaptic depolarizations recorded in S1 neurons upon M1 stimulations, which were effective in triggering APs in most of the cells with a high probability. Notably, firing latency of S1 neurons in response to M1 stimulations (~9 ms) shortly preceded the peak depolarization in MSNs, suggesting that a cumulative effect of converging, and temporally correlated, M1 and S1 CS inputs could intensify the fire rate of MSNs.

We found that striatal responsiveness to ipsilateral M1 stimulations was more reliable and robust compared to that consecutive to sensory stimuli. This could be due to multiple, and non-exclusive, mechanisms. First, intra-cortical electrical stimulations are likely to produce a massive recruitment of CS neurons, including a direct activation of M1 CS neurons and the subsequent synaptic excitation of S1 CS neurons, whereas sensory stimuli are conveyed to S1 CS neurons *via* a polysynaptic, and thus less reliable, network. Second, the ipsilateral CS projections from M1 into the dorsolateral striatum are more extensive than those originating from S1 [9], and thus more prone to contact a larger number of MSNs. And third, the synaptic efficacy of ipsilateral M1 CS neurons onto somatosensory MSNs seems to be greater compared to S1 CS connections, at least in the mouse dorsolateral striatum *in vitro* [22]. It has been previously reported that contralateral CS projections from S1 are much sparser than those originating from M1 [17, 49]. One can thus expect that the differences we observed between striatal responsiveness to ipsilateral S1 and M1 inputs be even more pronounced during activation of the contralateral cortices.

## Functional considerations

We demonstrated that MSNs located in the dorsolateral part of the striatum integrate converging excitatory synaptic inputs originating from whisker-related sensory and motor cortical areas, which are anatomically, and functionally, interconnected. These cortical areas are nearly co-activated during a sensorimotor task involving the whisker system [50], providing an optimal pattern of synaptic drive for MSNs which are known to operate as coincidence detector of

excitatory inputs [30, 31] because of their peculiar intrinsic electrical properties. Indeed, MSNs exhibit a powerful inwardly rectifying potassium selective current, responsible for most of their resting membrane conductance and highly polarized V$m$, and for a pronounced inward rectification during membrane hyperpolarization [31, 32, 37, 45]. As a consequence, the voltage-dependent activation of this potassium current is mainly accountable for the relatively low R$m$ and τ$m$ of MSNs when measured from the resting membrane potential. In addition, the presence of this potassium conductance in the dendritic tree of striatal neurons [51] likely causes a dramatic increase of the dendritic electrotonic length and, consequently, negatively alters the amplitude and shape of propagated synaptic depolarizations. This intrinsic control of membrane excitability and synaptic responses of MSNs acts in synergy with a feedforward inhibition caused by the activation of striatal fast-spiking GABAergic interneurons by CS afferents [19–22]. The firing of striatal interneurons produces a robust chloride-dependent postsynaptic conductance able to block the generation of APs in MSNs [52] and to shunt incoming excitatory synaptic inputs [13, 20].

According to these multiple mechanisms in the inhibitory control of MSNs excitability and responsiveness, it is hypothesized that only a large number of correlated excitatory inputs may be able to significantly depolarize the striatal membrane. Indeed, cortical synchronization is required to impede the feedforward inhibition of MSNs and to deactivate the inwardly rectifying potassium current, leading to an explosive change in the electrotonic structure of MSNs and an increased probability for $d$PSPs to reach voltage firing threshold [30, 31]. Our findings support such a dynamic scenario for striatal activation upon stimulation of M1. Hence, we have shown that electrical activation of M1 CS neurons engendered an initial membrane depolarization in MSNs, which is further, and rapidly, amplified by the fast intra-cortical recruitment of S1 CS neurons [26], which have dense overlapping inputs with M1 CS projections within the same striatal sector [5, 6, 9] and, as attested by the present study, onto the same MSNs. This augmented excitatory synaptic drive onto MSNs, which could also result from a fast sequential activation from S1 to M1 during sensory stimulation [53], may thus become sufficient to overcome the shunting effect caused by intrinsic membrane properties and the recruitment of GABAergic synaptic inputs, allowing MSNs to depolarize up to their firing threshold. This process is expected to induce a sustained synaptic activity and an increased firing rate in MSNs when the two converging cortical areas are concomitantly and repeatedly co-activated [41], ultimately leading to the optimization of striatum-related sensorimotor behaviors.

## Acknowledgments

We thank Sarah Lecas for help in neuronal reconstructions.

## Author Contributions

**Conceptualization:** Stéphane Charpier, Séverine Mahon.

**Formal analysis:** Stéphane Charpier, Morgane Pidoux.

**Investigation:** Stéphane Charpier, Morgane Pidoux.

**Supervision:** Séverine Mahon.

**Validation:** Stéphane Charpier, Séverine Mahon.

**Writing – original draft:** Stéphane Charpier, Séverine Mahon.

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
