## [Decision Letter · Decision Letter 0]

19 Nov 2019

PONE-D-19-30235

Converging sensory and motor cortical inputs onto the same striatal neurons: an in vivo intracellular investigation

PLOS ONE

Dear Pr. Charpier,

Thank you for submitting your manuscript to PLOS ONE. After careful consideration, we feel that it has merit but does not fully meet PLOS ONE’s publication criteria as it currently stands. Therefore, we invite you to submit a revised version of the manuscript that addresses the points raised during the review process.

We would appreciate receiving your revised manuscript by Jan 03 2020 11:59PM. To enhance the reproducibility of your results, we recommend that if applicable you deposit your laboratory protocols in protocols.io, where a protocol can be assigned its own identifier (DOI) such that it can be cited independently in the future. For instructions see: http://journals.plos.org/plosone/s/submission-guidelines#loc-laboratory-protocols

We look forward to receiving your revised manuscript.

Kind regards,

Jean-Pierre Mothet, Ph.D

Academic Editor

PLOS ONE

Journal Requirements:

Reviewers' comments:

Reviewer's Responses to Questions

**Comments to the Author**

1. Is the manuscript technically sound, and do the data support the conclusions?

Reviewer #1: Yes

Reviewer #2: Yes

Reviewer #3: Yes

2. Has the statistical analysis been performed appropriately and rigorously? 

Reviewer #1: Yes

Reviewer #2: Yes

Reviewer #3: Yes

3. Have the authors made all data underlying the findings in their manuscript fully available?

Reviewer #1: Yes

Reviewer #2: No

Reviewer #3: Yes

4. Is the manuscript presented in an intelligible fashion and written in standard English?

Reviewer #1: Yes

Reviewer #2: Yes

Reviewer #3: Yes

5. Review Comments to the Author

Reviewer #1: The paper by Charpier et al describes intracellular recordings from neurons in the dorsolateral striatum of anesthetized rats. The authors show that striatal projection neurons (MSNs) responding to contralateral whisker deflections also respond to electrical stimulation of the ipsilateral motor cortex, thus suggesting that they may be involved in sensorimotor processes. The authors also confirm that these responses to electrical stimulation are monosynaptic, based on the onset latencies in striatal and cortical neurons. The manuscript is interesting, timely, and well-written, and the data presented are convincing. I have just a few comments regarding additional information to be added and the relationship to previously published studies.

Sensorimotor targeting: The targeting of MSNs by sensory and motor cortices, even in overlapping regions, was shown in earlier papers using other methods: (Wall…Kreitzer, Neuron (2013), Reig & Silberberg, Cerebral cortex (2016), ). Moreover, it was shown that both types of MSNs (direct and indirect pathway neurons) receive these inputs. The issue of selective targeting of direct and indirect pathways should be addressed, especially as the data in the paper showing that a vast majority of recorded MSNs respond to both types of cortical inputs. These data strongly suggest that both MSN types respond to both types of cortical stimulation.

Was there any attempt to explain the diversity in response by staining for D1/D2/SP/ENK to determine dMSN or iMSN identity of recorded neurons?

Another comparison would be to assess the difference between S1 and M1 responses under the same conditions. Currently, the stimulation of S1 is done indirectly by whisker deflection while M1 is activated electrically. Do the authors have comparable data showing MSN responses to electrical stimulation in S1 compared to M1? If such electrical stimuli are given with same parameters, one could conclude whether MSNs in dorsolateral striatum receive stronger or weaker inputs from these regions.

Bilateral projections: previous studies (Brown et al., Neuroscience (1996); Lei et al, J.Neuroscience (2004); Reig and Silberberg, cerebral cortex (2016)) have shown that one of the main differences between corticostriatal projections from primary sensory and motor cortices is the density of contralateral innervation. Projections to contralateral striatum from primary sensory cortex are much sparser than those from motor cortex and other more frontal regions. Do the authors have any data regarding bilateral whisker and/or motor cortex stimulation? One interesting question that could be answered using the experimental setup is the comparison between ipsilateral and contralateral electrical and sensory stimulation. The prediction, based on previous data is that such differences will be more pronounced in the sensory responses than motor stimulation.

Feedforward inhibition: the role of inhibitory connections within the striatum (mainly striatal interneurons but also MSN collaterals) was shown to be instrumental in shaping both spontaneous activity and whisker evoked responses (Reig & Silberberg, neuron 2014). Moreover, inputs to striatal interneurons (in particular FS interneurons) were shown to arise from various cortical regions, including sensory, motor and others. This suggests that the responses described in this study (both sensory and electrical stimuli) are shaped by an inhibitory component as well. This issue should be addressed, at least in discussing the cortical inputs, and if possible also in presentation of experimental data. Such an inhibitory component could be revealed by depolarizing the MSNs recorded intracellularly.

Minor:

The authors show that 87% of recorded neurons respond to whisker stimulation. This is significantly higher than their earlier paper (2011) that showed 55% of responding neurons. What could be the reason for this difference in responsiveness? Was there any difference in the recording region or stimulus presentation?

The order of authors is different in the paper heading and in the cover pages (2nd and 3rd authors).

Results page 8: “…relatively weak Rm…”. Better to use high/low Rm.

Reviewer #2: In this manuscript, Charpier et al., investigated if projection neurons in the dorsolateral region of the striatum respond to both cortical input from functionally-related sensory and cortical regions. A key aspect of this study is that it is conducted in vivo, using intracellular electrophysiological recordings of striatal projection neurons (76 in total, which is a large number, considering the technical challenge of such approach), thus allowing to monitor supra et subthreshold changes in the membrane potential, in response to cortical/whisker stimulation. First, the authors reported that a large proportion of the striatal neurons recorded respond to whisker stimulation either through sub-threshold or supra-threshold depolarizing potentials. EcoG and intracell recordings in the barrel cortex provide strong evidence that whisker-evoked responses in the striatum were cortically driven. Then the authors showed that the same neurons also responded to electrical stimulation of M1 and provided evidence that these responses were mono-synaptically driven (rather than secondary to intra-cortical stimulation of S1). A key experiment was to record simultaneously S1 and striatal neurons while stimulating M1. The authors reported that striatal neurons responded with a shorter latency than S1 to M1 stimulation.

Altogether the authors provide excellent and rare in vivo electrophysiolgical data that demonstrate that the same striatal projection neurons can respond to sensory and motor cortical input from a functionally-related brain region. This knowledge is critical to understand the function of this brain region.

The study was carefully and rigorously conducted and the results which are in agreement with previous anatomical studies, obtained in vivo, constitute an important piece of information that should be shared with the community.

I have a few minor suggestions that the author may consider to address to improve their manuscript.

1) I was surprised by the opening sentence of the manuscript and the references cited. The authors mentioned the role of the striatum production and optimization of context-dependant behavior (isn't this the purpose of the whole nervous system?) and cite relatively “old” reviews of Ann Graybiel, which postulate a role of the striatum in motor control akin to as sort of traffic light signaling the beginning and end of automatized motor sequences, and thus overlook the moment to moment sensory representation associated with any behavior.

In my opinion, there are studies from the Silberberg and West labs that could have been provided a better context for this study to emphasis the importance of sensorimotor integration in the striatum (see also Robbe 2018 for a recent review on this topic)

2) A recent detailed anatomical tracing study nicely illustrate that vibrissal M1 and S1 send convergent input into specific region of the striatum (Hooks et al., 2018) and that this convergence does not occur for functionnaly distinct cortical area. This work should be cited

3) In the introduction, the authors suggested that sensory information is refined in the striatum. But from the description provided one could very well conclude that the information is degraded in the striatum. In fact both interpretation (refinement vs degradation or even filtering) could be correct. Maybe the author should use a more neutral word than “refinement” ?

4) When reading the result section for the first time it was not clear to me if the neurons recorded while stimulating M1 where the same one than those for which whisker responsiveness had been tested. The authors should make it very clear in the text (p10, after subsection title) that these are the same neurons.

5) The authors recorded 76 projection neurons. Most likely half of them where striatopallidal neurons while the other half. There has been anatomical evidence that striatonigral (direct-pathway) neurons are more strongly connected with the sensory cortex while striatopallidal (indirect-pathway) neurons are more connected with the motor cortex. It would have been nice if the authors would have speculated on the possible dichotomy of responses (or absence of) between these two populations.

6) I am not sure the authors should use a t-test to compare latencies between responses of striatal and S1 neurons (figure 5). The number of S1 is small (n=8) and I doubt normality can be evaluate. A non parametric test would be more appropriate, even if I have no doubt the result will be the same.

David Robbe

Reviewer #3: In this manuscript, the authors examine and characterize the cortico-striatal inputs from M1 and S1 to MSNs of the dorsolateral striatum (DLS) during whisker-induced sensory processing. Their major aim is to evaluate whether M1 and S1 inputs may converge on the very same MSN in this striatal area. The authors used an approach combining S1 and MSN intracellular recording and S1 electrocorticogram in vivo in anaesthetized rats subjected to whisker stimulation. The data showed an heterogenous response of DLS-somatosensory MSN to whisker sensory stimuli, and indirectly demonstrated a short-lasting direct monosynaptic M1 to MSN DLS connection. The data also demonstrated that sensory evoked whisker stimuli elicit responses in MSNs which are also responsive to M1 input, confirming their hypothesis of a M1 - S1 input convergence on individual MSNs in the DLS.

The question addressed in this paper is of high interest in the field to more deeply understand the motor and somatosensory input processing in the striatum. The experimental design providing in vivo recordings in DLS and the S1 cortex in a paradigm of sensory stimulation is highly adequate and perfectly conducted, and, the manuscript is clearly written with figures and text describing the results accurately. It delivers data that reinforce the knowledge on striatal computation of diverse cortical inputs. The following minor concerns must be considered.

The authors acknowledged the fact that CS inputs activation in their paradigms may or should result in both direct excitatory activation and feedforward inhibition of MSN, through activation of interneurons (see the discussion on the increase in synaptic conductance). However, no experiments were performed to evaluate the contribution of this feedforward inhibition in the protocols used in this study. Intrastriatal pharmacological GABAergic blockade could help to identify this contribution and this could be therefore clarified.

The data examining the membrane conductance of MSNs during the M1-induced response as illustrated in fig 4D should be more clearly presented. Indeed, the negative current pulse has a duration of 5ms in case of conjunction with the M1-induced responses (bottom traces) whilst it is of hundreds of ms in absence of stimulation (top trace). Moreover, calculation and comparison of conductances in both conditions should be provided.

6. PLOS authors have the option to publish the peer review history of their article (what does this mean?). If published, this will include your full peer review and any attached files.

Reviewer #1: No

Reviewer #2: Yes: David Robbe

Reviewer #3: No

---

## [Author Response · Author response to Decision Letter 0]

19 Dec 2019

Reviewer # 1 

We thank the reviewer for his/her positive assessment of this work and appreciate the suggestions, which have been carefully taken into account. 

Major points

1. “Sensorimotor targeting: The targeting of MSNs by sensory and motor cortices, even in overlapping regions, was shown in earlier papers using other methods: (Wall…Kreitzer, Neuron (2013), Reig & Silberberg, Cerebral cortex (2016)). Moreover, it was shown that both types of MSNs (direct and indirect pathway neurons) receive these inputs. The issue of selective targeting of direct and indirect pathways should be addressed, especially as the data in the paper showing that a vast majority of recorded MSNs respond to both types of cortical inputs. These data strongly suggest that both MSN types respond to both types of cortical stimulation.” “Was there any attempt to explain the diversity in response by staining for D1/D2/SP/ENK to determine dMSN or iMSN identity of recorded neurons?”

Answer: Our study, at this stage, was specifically designed to examine the CS sensorimotor convergences onto single MSNs, regardless their biochemical content and output connectivity. As pointed out by the Reviewer, it is very likely that the two subpopulations of MSNs (striatonigral and striatopallidal) were indeed included in the data set given the elevated number of recorded MSNs (n = 75). Our data showing that all recorded cells could respond to M1 stimuli and that the three classes of somatosensory MSNs, differentially responding to S1 inputs, did not exhibit specific intrinsic membrane properties (as it is expected for direct and indirect pathway striatal neurons) do not permit to conclude on a possible distinct processing of S1 and M1 inputs by the two subpopulations of MSNs. However, to follow the Referee’s suggestion, this crucial issue is now discussed (page 14, lines 9-14). 

2. “Currently, the stimulation of S1 is done indirectly by whisker stimulation while M1 is activated electrically. Do the authors have comparable data showing MSN responses to electrical stimulation in S1 compared to M1? If such electrical stimuli are given with same parameters, one could conclude whether MSNs in dorsolateral striatum receive stronger or weaker inputs from these regions.”

Answer: Contralateral air-puff stimulations were here preferred to electrical stimulations of the S1 cortex to activate somatosensory CS projections in a more physiological way. Moreover, this approach allowed assessing the impact of M1 CS inputs on the three classes of somatosensory MSNs we previously identified using natural sensory stimuli, i.e. dPSP-responding, AP-responding and non-responding MSNs. It is likely that electrical stimulations of S1, causing a massive non-specific activation of S1 CS neurons, would have precluded the distinction between the three types of somatosensory MSNs.

 Although our experimental design did not permit to compare the strength of S1 and M1 inputs onto striatal cells (a point which is discussed on page 15, lines 3-11), it led us to demonstrate that all the functional subpopulations of somatosensory MSNs could indeed be excited by CS motor inputs. 

3. “Bilateral projections: previous studies (Brown et al., Neuroscience (1996); Lei et al, J.Neuroscience (2004); Reig and Silberberg, cerebral cortex (2016)) have shown that one of the main differences between corticostriatal projections from primary sensory and motor cortices is the density of contralateral innervation. Projections to contralateral striatum from primary sensory cortex are much sparser than those from motor cortex and other more frontal regions. Do the authors have any data regarding bilateral whisker and/or motor cortex stimulation? One interesting question that could be answered using the experimental setup is the comparison between ipsilateral and contralateral electrical and sensory stimulation. The prediction, based on previous data is that such differences will be more pronounced in the sensory responses than motor stimulation.”

 Answer: The differential impact of sensory and motor contralateral CS innervation onto striatal responsiveness is indeed a very interesting question, although not directly addressed in the present study. Comparing bilateral sensory and motor integration at the level of individual MSNs would require an extensive set of additional experiments, which are beyond the scope of this work. This important issue is now exposed as a forward-looking view point in the Discussion and appropriate literature is cited (page 15, lines 11-14). 

4. “Feedforward inhibition: the role of inhibitory connections within the striatum (mainly striatal interneurons but also MSN collaterals) was shown to be instrumental in shaping both spontaneous activity and whisker evoked responses (Reig & Silberberg, neuron 2014). Moreover, inputs to striatal interneurons (in particular FS interneurons) were shown to arise from various cortical regions, including sensory, motor and others. This suggests that the responses described in this study (both sensory and electrical stimuli) are shaped by an inhibitory component as well. This issue should be addressed, at least in discussing the cortical inputs, and if possible also in presentation of experimental data. Such an inhibitory component could be revealed by depolarizing the MSNs recorded intracellularly.”

Answer: We agree with the Referee that feedforward and collateral recurrent inhibition of MSNs, stimulated by CS inputs, is of crucial importance to shape striatal responsiveness, as it may both participate to a conductance increase in MSNs and tune their synaptic and firing responses. We have explored this issue in our previous study characterizing the sensory-evoked responses in somatosensory MSNs (Pidoux et al., 2011). By the use of KCl-filled microelectrodes, displacing the equilibrium potential of chloride in MSNs, we found that sensory stimuli, in such condition, could evoke larger synaptic depolarizations in MSNs resulting in an increased relative number of AP-responding neurons, indicating the presence of a powerful Cl--dependent conductance. We also showed, by means of extracellular recordings of striatal GABAergic interneurons, that contralateral whisker sensory stimuli could induce repetitive discharges in striatal interneurons. Overall, this demonstrates that CS neurons activation by sensory stimuli resulted in a potent feedforward inhibition of MSNs intermingled with the direct CS-induced glutamatergic synaptic excitation. It is very likely that a similar process occurs in the case of M1 stimulations through the activation of S1 CS neurons (as previously demonstrated) but also directly from the activation of M1 CS neurons, which are known to connect striatal interneurons in the somatosensory striatum (Ramanathan et al., 2002; see also Lee et al., 2019). Thus, as suggested by the Referee, this hypothesis is now developed in the Discussion (page 15, lines 29-37 and page 16, lines 1-2). 

Minor points

1. “The authors show that 87% of recorded neurons respond to whisker stimulation. This is significantly higher than their earlier paper (2011) that showed 55% of responding neurons. What could be the reason for this difference in responsiveness? Was there any difference in the recording region or stimulus presentation?”

Answer: The proportion of MSNs responding to sensory stimulations is indeed more elevated in this study compared to our previous investigation. However, we confirmed here the three types of responses and the percentage of AP-responding cells was found to be similar between the two studies, further demonstrating a cell-dependent filtering of sensory information in the striatum. The partial discrepancy between the two investigations does not reflect different experimental conditions or neuronal properties, including anesthesia procedures, sites of recordings, stimulus features and intrinsic membrane properties. Instead, it may result from the higher number of neurons recorded in the present study: 76 vs 49 in the earlier one, which could have refined the statistical assessment of the three neuronal populations. Moreover, because CS glutamatergic connections display various forms of activity-dependent plasticity, it is plausible that the experimental groups used in these two studies were subject to various levels of whisker-related behavioral learning, differentially affecting the CS connectivity. Although speculative, this hypothesis deserves to be tested in the future. 

2. “The order of authors is different in the paper heading and in the cover pages (2nd and 3rd authors).”

Answer: We thank the Reviewer for having warned us on this. Séverine Mahon is the last author; this has been corrected.

3. “Results page 8: “…relatively weak Rm…”. Better to use high/low Rm.”

Answer: “Weak” has been replaced by “low” (page 8, line 11).

Reviewer # 2 

We are grateful to Dr David Robbe for his constructive and thoughtful comments. The revised version has been amended following his suggestions. 

Minor suggestions

1. “I was surprised by the opening sentence of the manuscript and the references cited. The authors mentioned the role of the striatum production and optimization of context-dependant behavior (isn't this the purpose of the whole nervous system?) and cite relatively “old” reviews of Ann Graybiel, which postulate a role of the striatum in motor control akin to as sort of traffic light signaling the beginning and end of automatized motor sequences, and thus overlook the moment to moment sensory representation associated with any behavior. In my opinion, there are studies from the Silberberg and West labs that could have been provided a better context for this study to emphasis the importance of sensorimotor integration in the striatum (see also Robbe 2018 for a recent review on this topic)”.

 Answer: We agree with the Reviewer that the role of the striatum in sensory integration was not sufficiently emphasized in the previous Introduction. This part of the manuscript has been rewritten and appropriate literature has been cited (page 3, lines 2-9).

2. “A recent detailed anatomical tracing study nicely illustrate that vibrissal M1 and S1 send convergent input into specific region of the striatum (Hooks et al., 2018) and that this convergence does not occur for functionally distinct cortical area. This work should be cited”

Answer: We thank the Referee for having pointed out this important piece of recent literature, which is now cited in the Introduction (page 3, lines 7-9) and throughout the manuscript.

3. “In the introduction, the authors suggested that sensory information is refined in the striatum. But from the description provided one could very well conclude that the information is degraded in the striatum. In fact both interpretation (refinement vs degradation or even filtering) could be correct. Maybe the author should use a more neutral word than “refinement” ?”

Answer: The term “refinement”, used in our previous article [Pidoux et al., J Physiol. (London) 2011] to qualify the variability of sensory responsiveness among MSNs, was chosen to stress that only a fraction of striatal neurons could be excited by a given whisker stimulus, a process that may allow selecting pertinent cortical information at the striatal level. As suggested by the referee, and to stay close to the original concept, the word “selection” is now used (page 3, line 18). 

4. “When reading the result section for the first time it was not clear to me if the neurons recorded while stimulating M1 where the same one than those for which whisker responsiveness had been tested. The authors should make it very clear in the text (p10, after subsection title) that these are the same neurons.”

Answer: Indeed, out of the 76 neurons intracellularly recorded in this study, 75 were challenged by both sensory and motor cortex stimulations (as shown in Fig. 4). To avoid any ambiguity, this is now clearly stated in the Results section (page 10, line 21-22).

5. “The authors recorded 76 projection neurons. Most likely half of them where striatopallidal neurons while the other half. There has been anatomical evidence that striatonigral (direct-pathway) neurons are more strongly connected with the sensory cortex while striatopallidal (indirect-pathway) neurons are more connected with the motor cortex. It would have been nice if the authors would have speculated on the possible dichotomy of responses (or absence of) between these two populations.”

Answer: Our study, at this stage, was specifically designed to examine the CS sensorimotor convergences onto single MSNs, regardless their biochemical content and output connectivity. As pointed out by the Reviewer, it is very likely that the two subpopulations of MSNs (striatonigral and striatopallidal) were indeed included in the data set given the elevated number of recorded MSNs (n = 75). Our data showing that all recorded cells could respond to M1 stimuli and that the three classes of somatosensory MSNs, differentially responding to S1 inputs, did not exhibit specific intrinsic membrane properties (as it is expected for direct and indirect pathway striatal neurons) do not permit to conclude on a possible distinct processing of S1 and M1 inputs by the two subpopulations of MSNs. However, to follow the Referee’s suggestion, this crucial issue is now discussed (page 14, lines 9-14).

6. “I am not sure the authors should use a t-test to compare latencies between responses of striatal and S1 neurons (figure 5). The number of S1 is small (n=8) and I doubt normality can be evaluate. A non-parametric test would be more appropriate, even if I have no doubt the result will be the same.”

Answer: To follow the pertinent suggestion of the Referee, we have now applied a non-parametric test (Mann-Whitney Rank Sum test) and confirmed the significant difference between the response latency of MSNs and S1 neurons to M1 stimulations (P = 0.03). The new test and corresponding statistical value are indicated (page 7, lines 21-22 and page 13, lines 4 and 30), and the figure 5 has been modified accordingly. 

Reviewer # 3 

Referee’ statement. “The question addressed in this paper is of high interest in the field to more deeply understand the motor and somatosensory input processing in the striatum. The experimental design providing in vivo recordings in DLS and the S1 cortex in a paradigm of sensory stimulation is highly adequate and perfectly conducted, and, the manuscript is clearly written with figures and text describing the results accurately. It delivers data that reinforce the knowledge on striatal computation of diverse cortical inputs.”

Comment: We deeply thank the Referee for his/her very positive comments on both our experimental strategy and findings. We are also grateful for his/her pertinent suggestions, which have all been taken into account. We have indeed done our best to design an in vivo paradigm allowing both the assessment of M1-induced responses in somatosensory MSNs and the estimate of the relative impact of M1 cortex and the related S1 region on the same striatal neurons. We hope that this study will stimulate further studies devoted to the elucidation of the functional effects of converging CS inputs onto single MSNs and output networks of basal ganglia. 

Minor concerns

1. “The authors acknowledged the fact that CS inputs activation in their paradigms may or should result in both direct excitatory activation and feedforward inhibition of MSN, through activation of interneurons (see the discussion on the increase in synaptic conductance). However, no experiments were performed to evaluate the contribution of this feedforward inhibition in the protocols used in this study. Intrastriatal pharmacological GABAergic blockade could help to identify this contribution and this could be therefore clarified.”

Answer: We fully agree with the Referee that the role of feedforward inhibition of MSNs through CS-induced excitation of striatal interneurons is of crucial importance. It may both participate to a conductance increase in MSNs and tune their synaptic and firing responses. We have explored this issue in our previous study characterizing the sensory-evoked responses in somatosensory MSNs (Pidoux et al., 2011). By the use of KCl-filled microelectrodes, displacing the equilibrium potential of chloride in MSNs, we found that sensory stimuli, in such condition, could evoke larger synaptic depolarizations in MSNs resulting in an increased relative number of AP-responding neurons, indicating the presence of a powerful Cl--dependent conductance. We also showed, by means of extracellular recordings of striatal GABAergic interneurons, that contralateral whisker sensory stimuli could induce repetitive discharges in striatal interneurons. Overall, this demonstrates that CS neurons activation by sensory stimuli resulted in a potent feedforward inhibition of MSNs intermingled with the direct CS-induced glutamatergic synaptic excitation. It is very likely that a similar process occurs in the case of M1 stimulations through the activation of S1 CS neurons (as previously demonstrated) but also directly from the activation of M1 CS neurons, which are known to connect striatal interneurons in the somatosensory striatum (Ramanathan et al., 2002; see also Lee et al., 2019). Thus, as suggested by the Referee, this hypothesis is now developed in the Discussion (page 15, lines 29-37 and page 16, lines 1-2). 

2. “The data examining the membrane conductance of MSNs during the M1-induced response as illustrated in fig 4D should be more clearly presented. Indeed, the negative current pulse has a duration of 5ms in case of conjunction with the M1-induced responses (bottom traces) whilst it is of hundreds of ms in absence of stimulation (top trace). Moreover, calculation and comparison of conductances in both conditions should be provided”

Answer: The Referee is right. It could indeed be problematic to compare conductance changes from current pulses having different duration, even if in the present case the collapse of current-induced Vm drop (�Vm) demonstrates a dramatic increase in membrane conductance at the peak of synaptic depolarizations. However, to follow the pertinent remark of the Referee, we have now compared the �Vm evoked by the same current pulses (5-20 ms of duration; -0.4 nA, n = 8 cells) at the summit of synaptic potentials and 120-160 ms after M1 stimulations, i.e. after termination of synaptic responses. However, since the duration of current pulses was mostly shorter than the capacitive charge time of the MSN membrane, we have calculated the �Vm instead of Rm values. This is now illustrated in the new panel D of Figure 4 and the corresponding data, demonstrating a breakdown in the membrane resistivity during M1 stimulations, are given in the Results section (page 11, lines 34-37 and page 12, lines 1-5). The figure legend has been modified accordingly (page 11, lines 5-9).

---

## [Decision Letter · Decision Letter 1]

13 Jan 2020

Converging sensory and motor cortical inputs onto the same striatal neurons: an in vivo intracellular investigation

PONE-D-19-30235R1

Dear Dr. Charpier,

We are pleased to inform you that your manuscript has been judged scientifically suitable for publication and will be formally accepted for publication once it complies with all outstanding technical requirements.

With kind regards,

Jean-Pierre Mothet, Ph.D

Academic Editor

PLOS ONE

Additional Editor Comments (optional):

Reviewers' comments:

Reviewer's Responses to Questions

**Comments to the Author**

1. If the authors have adequately addressed your comments raised in a previous round of review and you feel that this manuscript is now acceptable for publication, you may indicate that here to bypass the “Comments to the Author” section, enter your conflict of interest statement in the “Confidential to Editor” section, and submit your "Accept" recommendation.

Reviewer #1: All comments have been addressed

Reviewer #2: All comments have been addressed

Reviewer #3: All comments have been addressed

2. Is the manuscript technically sound, and do the data support the conclusions?

Reviewer #1: Yes

Reviewer #2: Yes

Reviewer #3: Yes

3. Has the statistical analysis been performed appropriately and rigorously? 

Reviewer #1: Yes

Reviewer #2: Yes

Reviewer #3: Yes

4. Have the authors made all data underlying the findings in their manuscript fully available?

Reviewer #1: Yes

Reviewer #2: No

Reviewer #3: Yes

5. Is the manuscript presented in an intelligible fashion and written in standard English?

Reviewer #1: Yes

Reviewer #2: Yes

Reviewer #3: Yes

6. Review Comments to the Author

Reviewer #1: The authors have addressed my comments in the discussion section of the paper. No additional data has been added but I hope will be done in future studies.

Reviewer #2: (No Response)

Reviewer #3: (No Response)

7. PLOS authors have the option to publish the peer review history of their article (what does this mean?). If published, this will include your full peer review and any attached files.

Reviewer #1: No

Reviewer #2: Yes: David Robbe

Reviewer #3: No

---

## [Editor Report · Acceptance letter]

21 Jan 2020

PONE-D-19-30235R1 

Converging sensory and motor cortical inputs onto the same striatal neurons: an in vivo intracellular investigation 

Dear Dr. Charpier:

I am pleased to inform you that your manuscript has been deemed suitable for publication in PLOS ONE. Congratulations! Your manuscript is now with our production department. 

With kind regards,

on behalf of

Dr Jean-Pierre Mothet 

Academic Editor

PLOS ONE